# Identification of Wheat Germplasm Resistance to Late Sowing

Samia Mahgoub Omer Basheir [1,2,3,4,5], Yi Hong [1,2,3,4], Chao Lv [1,2,3,4], Hongwei Xu [6], Juan Zhu [1,2,3,4], Baojian Guo [1,2,3,4], Feifei Wang [1,2,3,4] and Rugen Xu [1,2,3,4,*]

1  Key Laboratory of Plant Functional Genomics of the Ministry of Education, Yangzhou University, Yangzhou 225009, China
2  Jiangsu Key Laboratory of Crop Genomics and Molecular Breeding, Yangzhou University, Yangzhou 225009, China
3  Jiangsu Co-Innovation Center for Modern Production Technology of Grain Crops, Yangzhou University, Yangzhou 225009, China
4  Joint International Research Laboratory of Agriculture and Agri-Product Safety of Ministry of Education of China, Yangzhou 225009, China
5  Faculty of Forestry, University of Khartoum, Shambat, Khartoum North 13314, Sudan
6  Shanghai Key Laboratory of Agricultural Genetics and Breeding, Biotechnology Research Institute, Shanghai Academy of Agricultural Sciences, Shanghai 201106, China
*  Correspondence: rgxu@yzu.edu.cn

**Abstract:** To evaluate the performance of wheat plant height and spike-related traits under delayed sowing conditions, a screening trial was conducted for wheat germplasm that exhibits resistance to late sowing and early maturity. The differences and stability of plant height and spike-related traits under different sowing dates were analyzed using 327 wheat germplasm sources from a wide range of areas. The results showed that mean values of wheat plant height and spike-related traits generally decreased along with the delay in sowing dates. Broad-sense heritability of plant height (PH), internode length below spike (ILBS), spike length (SL), spikelet per spike (SPS), and spike number (SN) under multiple environments were all above 85%. Ten varieties, including Xiangmai 35, Pingyang 27, Huaimai 23, Huaimai 22, Emai 6, Zhenmai 12, Xiaoyan 81, Shannong 7859, Annong 1589, and Shuiyuan 86 indicated stable performance under different sowing dates, which harboring good resistance to late sowing. The results of this study laid a foundation for breeding high-yield wheat varieties that are resistant to late sowing.

**Keywords:** wheat; sowing date; plant height; spike related traits; stable performance

## 1. Introduction

Wheat (*Triticum aestivum* L.) is one of the major staple food crops worldwide [1]. However, due to climate change and other factors, the yield and quality of wheat are expected to face significant challenges. Recent research indicates that global temperatures are increasing per decade at a rate of 0.18 °C, and it is estimated that each degree of temperature rise is estimated to reduce wheat production by 6% [2]. The main challenge for researchers is to increase yield and yield stability with minimum investment in the region. China is one of the largest wheat producers globally, accounting for 17% of the world's production [3]. Wheat varieties are generally divided into winter wheat and spring wheat, depending on their requirement for long-term cold exposure before flowering [4]. Winter wheat is one of the major crops in the middle and lower reaches of the Yangtze River, which is one of the major wheat-producing areas in southern China, and follows rice-wheat rotations farming system [5].

Yield is a complex, polygenic, and quantitative trait that is composed of multiple elements, including effective tiller number, grain number per spike, and grain weight. Yield is affected by many environmental factors [6]. One important agricultural management strategy for optimizing final yield is selecting an appropriate sowing date. Delayed sowing

dates can have a detrimental effect on yield, yield components, and other aspects of the growth and development of wheat plants [7]. This delay can affect the heading stage, filling stage, and maturity [8], leading to decreased number of spikes per plant [9] and yield per unit area. It is worth noting that these effects may vary across different varieties [10].

The timing of sowing has a notable impact on the ability of winter wheat to withstand cold temperatures. Early sowing promotes rapid growth and development, which can cause the vernalization stage to end early and move to the reproductive stage (jointing stage). At the jointing stage, when affected by low-temperature cold waves, the main stems and tillers are suffered freezing damage, leading to tiller death. In contrast, late sowing leads to late seedling emergence, weak seedlings, less photosynthate, and less stored sugar, which can result in freezing damage [11,12]. Late sowing can also shorten the fertility period of wheat and impact the development of important agronomic traits such as the number of spikes per plant, number of grains per spike, and grain weight, resulting in lower yields [13]. In recent years, the extension of rice growth periods and the frequent occurrence of rainy weather during the normal sowing date led to a general delay in wheat planting [14]. In the context of global warming trends, higher prewinter temperatures can lead to the overgrowth of wheat seedlings and a significant decrease in growth duration [15]. If low temperatures are experienced, wheat seedlings may suffer frost damage, which will also lead to wheat yield reduction [16,17]. Therefore, it is important to identify and screen late-sowing tolerant wheat varieties to mitigate the difficulties associated with the late-sowing practice.

Using 327 wheat germplasms, this study analyzed the impact of late sowing on wheat plant height and spike-related characteristics. Comprehensive evaluation and stability analysis was used to screen for varieties that exhibited good tolerance to late sowing. The aim of this study was to establish a crucial theoretical foundation for the enhancement and utilization of germplasm resources.

## 2. Materials and Methods

### 2.1. Test Materials

A total of 327 wheat germplasm collected from different regions were used to evaluate the impact of late sowing on plant growth and development. These materials include domestic wheat germplasm and indigenous wheat varieties (lines) grown in the middle and lower reaches of the Yangtze River and Huang-Huai-Hai region of china. The list of the germplasm is shown in Table S1.

### 2.2. Field Design

In the autumn of 2018, 327 accessions were planted in Yangzhou, China (YZ) with three sowing dates: October 27, November 10, and November 24 (stages I, II, and III), respectively, and in Yancheng, China (YC) on October 29, November 12, and November 26, respectively. Twelve seeds of each cultivar were planted 5 cm apart with 30 cm between rows with three replications.

### 2.3. Phenotypic Evaluation

After maturity, five plants from each row were randomly selected for phenotypic measurement, including plant height (PH), spike length (SL), internode length below spike (ILBS), spikelet per spike (SPS), and spike number (SN).

### 2.4. Descriptive Statistics

Maximum, minimum, and mean values and the coefficient of variation of each trait from different locations and sowing dates were analyzed using Microsoft Excel 2019 (Microsoft, Redmond, WA, 2019). The R package "lme4" was used to calculate broad-sense heritability for plant height and spike-related traits, and the best linear unbiased prediction (BLUP) of each trait under each sowing date was obtained by combining data from two locations [14]. IBM SPSS Statistics 21 (IBM SPSS Statistics 21; IBM, USA) was used to analyze the phenotype data, such as ANOVA and correlation analysis.

### 2.5. Genetic Diversity Analysis

Calculation of genetic diversity index ($H'$, Shannon-Wiener diversity index [18]): By calculating the mean ($\mu$) and standard deviation ($\sigma$) of tested materials, each trait was divided into 10 grades ($[xi < (\mu - 2\sigma)] - [xi > (\mu + 2\sigma)]$) (xi is the phenotype of the i-th variety) with every $0.5\sigma$ as a level. The relative frequency Pi of each level was calculated to obtain the genetic diversity index. The derived formula is: $H' = -\sum p_i \ln p_i$ ($p_i$ is the percentage of materials in grade i to the total number of materials).

### 2.6. Cluster Analysis

The K-means clustering method was used to cluster all wheat plant height and spike-related traits measured in this study. To ensure the optimal number of clusters, the gap statistic method was used as an essential step prior to clustering. The optimal number of clusters shows a larger gap statistic, which refers to the lower intracluster variation away from reference [19]. All analyses were performed in the R program (R Core Team, 2018).

### 2.7. Stability Analysis and Comprehensive Evaluation

AMMI model is a mathematical model with additivity and multiplicativity components, which combines analysis of variance and principal component analysis. It has become a hot topic to use the AMMI model to reveal the stability of varieties on yield traits by analyzing interactions between genotypes and locations [20,21]. The formulas of the model are calculated as follows:

$$y_{ijk} = \mu + \alpha_i + \beta_j + \sum_{r=1}^{n} \theta_r \gamma_{ir} \delta_{jr} + \rho_{ij} + \varepsilon_{ijk}$$

$$D_i = \sqrt{\sum_{r=1}^{N} W_r \gamma_{ir}^2} \qquad D_j = \sqrt{\sum_{r=1}^{N} W_r \delta_{jr}^2}$$

In these formulas: $y_{ijk}$ is the k-th repeat phenotype value of trait of the i-th variety (line) in the j-th environment. $\mu$ represents the average trait phenotype. $\alpha_i$ represents the main effects of the i-th variety (genotype), $\beta_j$ represents the main effects of the j-th environment. $\theta_r$ represents the eigenvalue of the r-th principal component of genotype and environment interactions (IPCA). $\gamma_{ir}$ represents the genotype score of the r-th IPCA. $\delta_{jr}$ represents the environment score of the r-th IPCA. $\rho_{ij}$ and $\delta_{jr}$ represent the residual and error, respectively. $D_i$ and $D_j$ represent the stability of varieties and the discrimination of environments, respectively. The lower $D_i$, the higher stability of varieties. The higher the $D_j$ value, the higher the discriminative power of environments on the variety. $W_r$ is the percentage of the variation explained by each IPCA, that is, the weight.

To calculate the overall preference for the stability of each material under delayed sown conditions, the Technique for Order of Preference by Similarity to the Ideal Solution (TOPSIS) was performed as a scientific evaluation method [22]. The relevant calculation includes the following formula:

$$y_{ij} = \frac{x_{ij} - \min\limits_{1 \leq i \leq n}(x_{ij})}{\max\limits_{1 \leq i \leq n}(x_{ij}) - \min\limits_{1 \leq i \leq n}(x_{ij})} \qquad y_{ij} = \frac{\max\limits_{1 \leq i \leq n}(x_{ij}) - x_{ij}}{\max\limits_{1 \leq i \leq n}(x_{ij}) - \min\limits_{1 \leq i \leq n}(x_{ij})}$$

$$z_{ij} = \frac{y_{ij}}{\sqrt{\sum_{i=1}^{n} y_{ij}^2}} \qquad p_{ij} = \frac{z_{ij}}{\sum_{i=1}^{n} z_{ij}}$$

$$e_j = -\frac{1}{\ln n} \sum_{i=1}^{n} p_{ij} \ln(p_{ij}) \qquad W_j = \frac{1 - e_j}{1 - \sum_{j=1}^{m}(1 - e_j)}$$

$$D_i^+ = \sqrt{\sum_{j=1}^{m}\left(W_jp_{ij} - \max_{1\le i\le n}\left(W_jp_{ij}\right)\right)^2} \quad D_i^- = \sqrt{\sum_{j=1}^{m}\left(W_jp_{ij} - \min_{1\le i\le n}\left(W_jp_{ij}\right)\right)^2}$$

$$S_i = \frac{D_i^-}{D_i^+ + D_i^-}$$

In these formulas: $y_{ij}$ represents the j-th trait phenotype of the i-th variety after positively formulated. $z_{ij}$ represents the standardization of phenotype. $p_{ij}$ represents the proportion of the j-th trait of the i-th variety in all varieties. $e_j$ represents the entropy of the j-th trait. $W_j$ represents a proportion of the j-th trait among all traits (weight). $D_i^+$ and $D_i^-$ denote the Euclidean distance between the phenotype and the maximum, and the phenotype and the minimum value of each trait in the i-th variety, respectively. $S_i$ denotes the comprehensive score of the i-th variety.

## 3. Results

### 3.1. Performance of Plant Height Traits and Spike Related Traits in Wheat

Table 1 shows the performance of wheat plant height and spike-related traits across various locations and sowing dates. In Yangzhou, the PH ranged from 53 to 189 cm for the first sowing date, 49 to 165 cm for the second sowing date, and 46 to 172 cm for the third sowing date. Meanwhile, the ILBS varied between 18 to 59 cm, 17 to 62 cm, and 16 to 61 cm for the first, second, and third sowing dates, respectively. The SL of the wheat population ranged between 7 to 17 cm, 7 to 16 cm, and 6 to 17 cm for the first, second, and third sowing dates, respectively. Similarly, the SPS ranged from 15 to 25, 16 to 25, and 15 to 25 for the first, second, and third sowing dates, respectively, while SN ranged from 4 to 24, 5 to 19, and 4 to 16 for the same sowing dates. In Yancheng, the PH of the wheat population varied from 56 to 165 cm, 53 to 160 cm, and 50 to 159 cm for the first, second, and third sowing dates, respectively, while the ILBS ranged from 17 to 58 cm, 16 to 58 cm, and 13 to 66 cm. The SL varied from 7 to 17 cm, 6 to 16 cm, and 6 to 16 cm, while the SPS ranged from 14 to 24, 15 to 35, and 15 to 25 for the first, second, and third sowing dates, respectively. The SN for Yancheng was between 4 to 15, 4 to 13, and 4 to 15 for the first, second, and third sowing dates, respectively.

**Table 1.** Performance of various traits under different locations and sowing dates.

| Location | Trait | Stage I | | | Stage II | | | Stage III | | |
|---|---|---|---|---|---|---|---|---|---|---|
| | | Mean | Range | CV | Mean | Range | CV | Mean | Range | CV |
| | PH (cm) | 108 | 53–189 | 28.71% | 101 | 49–165 | 30.68% | 99 | 46–172 | 31.01% |
| | ILBS (cm) | 36 | 18–59 | 28.48% | 36 | 17–62 | 31.18% | 35 | 16–61 | 29.32% |
| YZ | SL (cm) | 10 | 7–17 | 15.09% | 10 | 7–16 | 14.88% | 10 | 6–17 | 15.32% |
| | SPS | 20 | 15–25 | 8.63% | 20 | 16–25 | 7.68% | 19 | 15–25 | 7.88% |
| | SN | 9 | 4–24 | 28.51% | 9 | 5–19 | 28.04% | 8 | 4–16 | 24.19% |
| | PH (cm) | 107 | 56–165 | 24.34% | 101 | 53–160 | 24.46% | 91 | 50–159 | 26.75% |
| | ILBS (cm) | 35 | 17–58 | 24.35% | 33 | 16–58 | 25.68% | 34 | 13–66 | 29.38% |
| YC | SL (cm) | 11 | 7–17 | 14.81% | 10 | 6–16 | 15.17% | 10 | 6–16 | 16.99% |
| | SPS | 19 | 14–24 | 8.90% | 20 | 15–35 | 8.94% | 19 | 15–25 | 7.84% |
| | SN | 7 | 4–15 | 30.79% | 7 | 4–13 | 25.73% | 6 | 4–15 | 30.38% |

Notes: YZ represents Yangzhou; YC represents Yancheng; PH represents plant height; ILBS represents internode length below spike; SL represents spike length; SPS represents spikelet per spike; SN represents spike number; CV represents the coefficient of variation; Stage I, II, and III represent different sowing dates, respectively.

The coefficient of variation of SPS of the wheat population was the lowest compared with other traits under different locations and sowing dates. At the same time, PH, ILBS, and SN had larger coefficients of variation. In Yangzhou, the mean values of all traits in the wheat population showed the same trend during the sowing period, with the mean values gradually decreasing with the delay of the sowing date. In Yancheng, the average values

of the three traits were the highest at the second sowing date, while PH and SL gradually decreased with the delayed sowing dates. The heritability of each trait was 98.45% for PH, 97.08% for ILBS spike, 95.04% for SL, 91.34% for SPS, and 87.61% for SN, respectively. PH and IBLS showed no significant interactions between varieties and sowing dates, as well as the spikelet per spike between locations, significant or extremely significant differences were observed between varieties, locations, sowing dates, interactions of varieties and locations, and interactions of varieties and sowing date in the rest characters, indicating that wheat plant height and spike related traits belong to the typical quantitative traits that controlled by multiple genes (Table 2).

**Table 2.** Variance analysis of wheat plant height traits and spike-related traits.

| SOV | Df | PH | | ILBS | | SL | | SPS | | SN | |
|---|---|---|---|---|---|---|---|---|---|---|---|
| | | MS | F | MS | F | MS | F | MS | F | MS | F |
| Varieties | 326 | 4402.54 | 84.18 ** | 508.43 | 42.15 ** | 11.78 | 26.18 ** | 10.88 | 11.89 ** | 18.01 | 8.87 ** |
| Environments | 1 | 5229.55 | 99.99 ** | 996.03 | 82.56 ** | 11.23 | 24.96 ** | 0.87 | 0.95 | 2039.4 | 1004.82 ** |
| Sowing dates | 2 | 25,149.87 | 480.86 ** | 106.36 | 8.82 ** | 69.4 | 154.19 ** | 168.94 | 184.63 ** | 231.76 | 114.19 ** |
| Varieties × Environments | 326 | 168.8 | 3.23 ** | 30.2 | 2.5 ** | 1 | 2.22 ** | 1.11 | 1.22 * | 2.74 | 1.35 ** |
| Varieties × Sowing dates | 652 | 44.42 | 0.85 | 10.41 | 0.86 | 0.55 | 1.22 ** | 1.05 | 1.14 * | 2.32 | 1.15 * |
| Error | 654 | 52.3 | | 12.06 | | 0.45 | | 0.92 | | 2.03 | |

Notes: * and ** indicate significant differences at the 0.05 and 0.01 levels, respectively. PH represents plant height; ILBS represents internode length below spike; SL represents spike length; SPS represents spikelet per spike; SN represents spike number; SOV represents sources of variation; Df represents degrees of freedom; MS represents the mean square.

*3.2. Correlation Analysis of Plant Height Traits and Spike-Related Traits in Wheat at Different Sowing Dates*

PH showed a positive correlation with ILBS under varying sowing dates, with correlation coefficients of 0.87, 0.93, and 0.94 for the three sowing dates, respectively. In contrast, the correlation coefficients between PH and SL were relatively low, being 0.25, 0.42, and 0.46, respectively. Similarly, the correlation coefficients between PH and SN were 0.37, 0.43, and 0.56, respectively. ILBS was positively correlated with SL, with correlation coefficients of 0.29, 0.40, and 0.42, respectively. SN was also positively correlated with ILBS, with correlation coefficients of 0.25, 0.41, and 0.51, respectively. SPS exhibited a positive correlation with SL, with correlation coefficients of 0.22, 0.23, and 0.38 for the first, second, and third sowing dates, respectively. Notably, the correlation coefficient between SL and SN changed from insignificant under the first sowing date to significant under the second (0.12) and third sowing dates (0.14). Under the third sowing date, the correlation of PH with ILBS and SPS was significant, measuring 0.19 and 0.16, respectively (Figure 1A).

*3.3. Genetic Diversity Analysis between Plant Height Traits and Spike-Related Traits in Wheat at Different Sowing Dates*

Under different sowing dates, the genetic diversity index of SL was the highest, and PH genetic diversity index was the lowest (Table 3). The distribution of each character was symmetrical, mainly concentrated in the third to eighth grades (Figure 1B–D). Except for PH, ILBS and SN, which were not distributed in the first grade, the other traits were distributed in all 10 grades. With the change of sowing date, the highest distribution grade of SL, SPS, and SN changed, while the highest distribution grade of PH and ILBS were relatively stable.

**Table 3.** Genetic Diversity Index of wheat traits at different sowing dates (*H'*).

| Trait | Stage I | Stage II | Stage III |
|---|---|---|---|
| PH | 1.8384 | 1.8703 | 1.8875 |
| ILBS | 2.0339 | 2.0082 | 1.9891 |
| SL | 2.0463 | 2.0084 | 2.0089 |
| SPS | 2.0245 | 1.9928 | 1.9419 |
| SN | 1.9534 | 2.0003 | 1.9867 |

Notes: PH represents plant height; ILBS represents internode length below spike; SL represents spike length; SPS represents spikelet per spike; SN represents spike number; Stage I, II, and III represent different sowing dates, respectively.

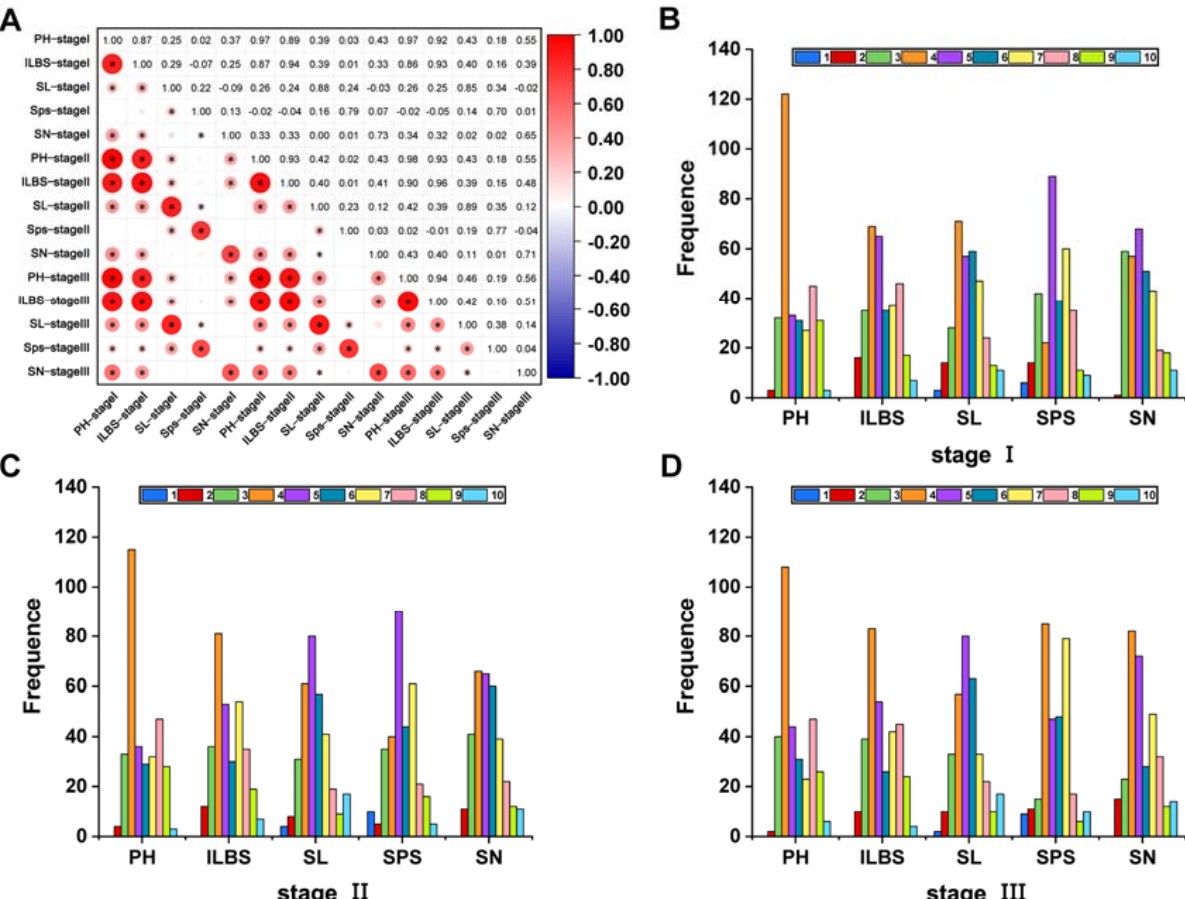

**Figure 1.** Correlation and genetic diversity analysis under different sowing dates. (**A**) Correlation analysis among wheat plant height and spike-related traits. (**B**) Grades distribution of each trait under the first sowing date. (**C**) Grades distribution of each trait under the second sowing date. (**D**) Grades distribution of each trait under the third sowing date. Different colors represent different grades from 1 to 10. Stage I, II, and III represent different sowing dates, respectively. YZ represents Yangzhou; YC represents Yancheng; PH represents plant height; ILBS represents internode length below spike; SL represents spike length; SPS represents spikelet per spike; SN represents spike number. * indicate significant differences at the 0.05 level.

*3.4. Phenotypic Clustering of Plant Height Traits and Spike-Related Traits in a Wheat Population*

To determine the optimal K number of the population, gap statistic was introduced as the judgment basis. The results are shown in Figure 2A. When K = 2, there was an obvious inflection point, and with the gradual increase of K, the gap statistic rose smoothly. Therefore, the population can be divided into two types according to the phenotypic value of each trait.

The k-means clustering result of tested wheat germplasm at K = 2 are plotted in Figure 2B, and the mean values of each trait of different categories are listed in Table 4. The results showed that the main composition 1 and 2 explain 72.67% of the total variation information. Subgroup 1 contained 203 materials, mainly from Jiangsu (70), Henan (32), and Shandong (23), and most of them were bred cultivars. These materials generally had lower plant height, internode length below spike, spike length and spike number. Subgroup 2 contained 124 materials, which has a wide geographical distribution with many landraces. The plant height and internode length below the spike were larger in this subgroup.

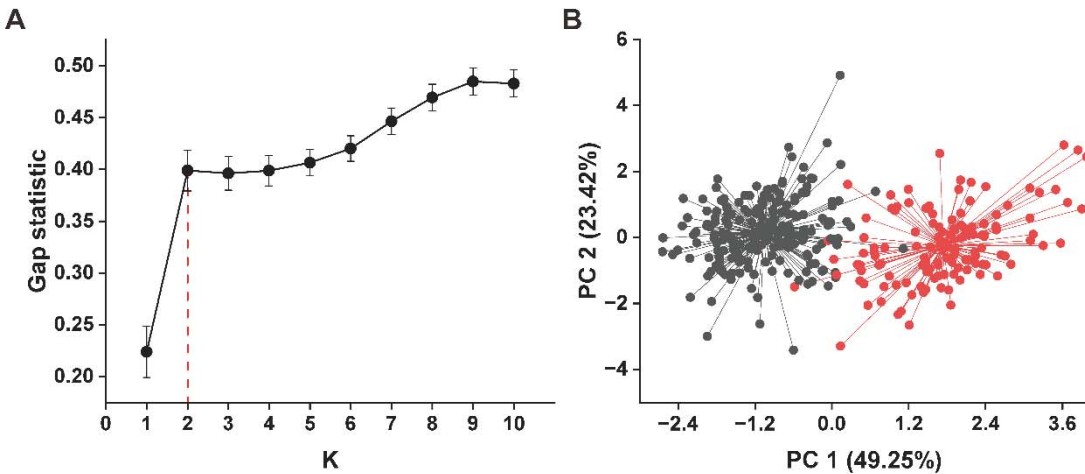

**Figure 2.** Cluster analysis of plant height and spike-related traits. (**A**) Gap statistic value of each cluster group. The red dashed line represents the optimal K number. (**B**) Results of K-means clustering. The grey and red represent subgroup 1 and subgroup 2, respectively.

**Table 4.** The performance of traits in different wheat categories.

| Subgroup | PH (mm) | ILBS (mm) | SL (mm) | SPS | SN |
|---|---|---|---|---|---|
| 1 | 82 a | 29 a | 10 a | 19 | 7 a |
| 2 | 132 b | 45 b | 11 b | 19 | 9 b |

Notes: PH represents plant height; ILBS represents internode length below spike; SL represents spike length; SPS represents spikelet per spike; SN represents spike number; Different letters indicate significant differences at 0.01 levels.

*3.5. Stability Analysis and Comprehensive Evaluation of Plant Height Traits and Spike-Related Traits in Wheat*

As shown in Table 5, the top five principal components of PH, ILBS, and SL, the top four principal components of SPS, and the top three principal components of SN reached a significant level. According to the AMMI model biplot (Figure 3), different environments had different discriminative power on wheat plant height and spike-related traits. Among them, three sowing dates in the Yangzhou pilot environment all had large Dj values, indicating that Yangzhou had the highest discriminative power on the stability of each trait of this wheat population.

**Table 5.** Principal component analysis of genotypes-by-environments (*F* value).

| IPCA | Df | PH | ILBS | SL | SPS | SN |
|---|---|---|---|---|---|---|
| IPCA1 | 330 | 32.93 | 36.96 | 14.54 | 2.50 | 2.02 |
| IPCA2 | 328 | 16.43 | 16.76 | 12.33 | 1.84 | 1.81 |
| IPCA3 | 326 | 9.75 | 12.80 | 10.66 | 1.64 | 1.24 |
| IPCA4 | 324 | 4.91 | 11.84 | 4.65 | 1.26 | |
| IPCA5 | 322 | 2.38 | 5.29 | 2.00 | | |

Notes: PH represents plant height; ILBS represents internode length below spike; SL represents spike length; SPS represents spikelet per spike; SN represents spike number; Df represents degrees of freedom; IPCA represents interaction principal component axes.

The stability parameters Di of traits were calculated according to the significant IPCA score (Table S1). Among all germplasms, the top five with better PH stability included Gaojiasuo (0.0770), Rosella (0.0948), Yangmai 3 (0.1086), Shannong 7859 (0.1113), and Bao205 (0.1136). The bottom five germplasm with poor stability included 77-M94 (2.7270), Nonglin 46 (2.8490), 980-1 (2.9971), 77-M63.64 (3.0973), and 980-2 (3.7595). The top five germplasm with better stability of ILBS included Zhoumai 42 (0.0431), Shannong 7859 (0.0462), Luo6073 (0.0659), Jimai 22 (0.0660), and Annong 0711 (0.0727). The bottom five germplasm with poor stability included Jinmai 33 (1.3375), Hongmangwugongjiao (1.3680),

Hongshimai (1.3863), Nonglin 46 (1.7603), and Neixiang5 (1.8175). The top five germplasm with better SL stability included Zhengmai 32 (0.0178), Emai 6 (0.0332), Yangmai 6 (0.0369), Zangdong 4 (0.0401), and Huaimai 23 (0.0405). The bottom five germplasm with poor stability included Nonglin 46 (0.5312), 980-1 (0.5731), Baiguanmai (0.5871), Hongmang-caizihuang (0.6008), and Baimangzaoxiaomai (0.9176). The top five germplasm with better stability in the number of main SPS included red husk Chikewumang (0.0246), Yangmai 19 (0.0328), Yangmai 20 (0.0371), Yangmai 23 (0.0417), and Zangdong 4 (0.0492), The bottom five germplasm with poor stability included Meiqianwu (0.4450), Xichangzao (0.4576), Mexican (0.5023), Villa Glori (0.6697), and Lumai 1 (1.2748). The top five germplasm with better SN stability included Gui79 (0.0045), Hongmixian (0.0120), Zhenmai 12 (0.0180), Xiaoyan 81 (0.0183), and Bimai 26 (0.0287). The bottom five germplasm with poor stability included Dianxihongkeyangmai (0.7852), Zhongguochun (0.7975), Nongda 183 (0.8503), Dezhou 845,153 (0.8722), and Sanyuehuang (0.8909).

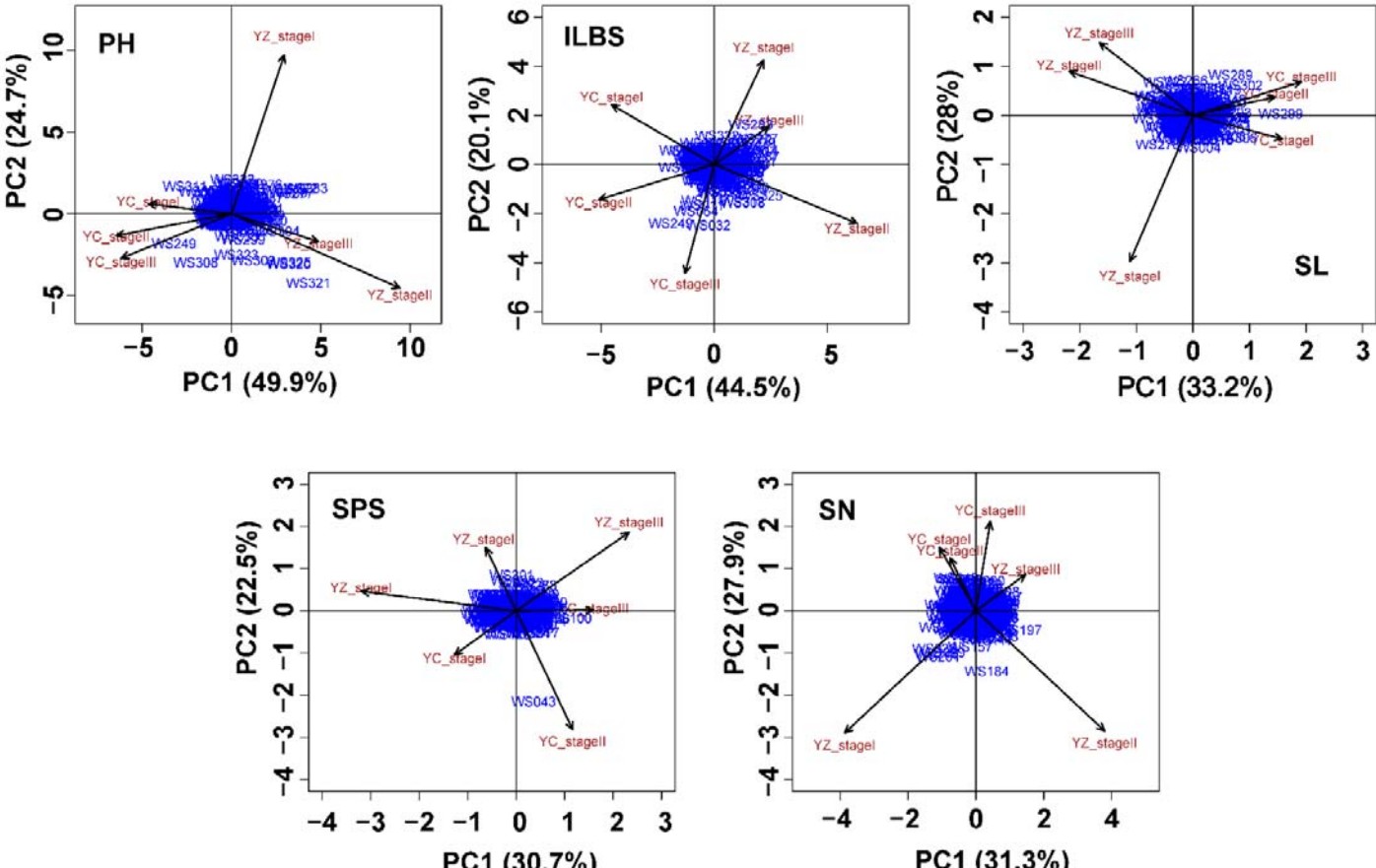

**Figure 3.** AMMI biplot of height and spike-related traits in the population. Blue labels represent the position of each variety (line); the closer the position to the origin, the lower the Di values. Black arrows represent the distance between each environment and origin; the longer arrows, the higher the Dj values. Stage I, II, and III represent different sowing dates, respectively. YZ represents Yangzhou; YC represents Yancheng; PH represents plant height; ILBS represents internode length below spike; SL represents spike length; SPS represents spikelet per spike; SN represents spike number.

The weight values of PH, ILBS, SL, SPS, and SN were 16.89%, 30.05%, 16.79%, 7.18%, and 29.10%, respectively. The comprehensive scores of each variety (line) are shown in Table S1. The results showed that the top 10 accessions with higher comprehensive scores were Xiangmai 35 (0.9336), Pingyang 27 (0.9325), Huaimai 23 (0.9258), Huaimai 22 (0.9224), Emai 6 (0.9216), Zhenmai 12 (0.9214), Xiaoyan 81 (0.9208), Shannong 7859 (0.9178), Annong

1589 (0.9176), and Shuiyuan 86 (0.9156). These accessions are mainly from the middle and lower reaches of the Yangtze River and Huanghuai.

## 4. Discussion

### 4.1. Effects of Sowing Date on Wheat Plant Height Traits and Spike-Related Traits

The timing of sowing is a critical factor that impacts the cold resistance of wheat. As the crop grows, different growth stages exhibit varying degrees of sensitivity to low temperatures. If wheat is sown as per the schedule and then subjected to spring cold, it may result in freeze injury to wheat seedlings that have experienced high accumulated temperature and excessive growth before winter. This can ultimately lead to reduced production. Conversely, selecting a suitable sowing date can help maximize the use of temperature and water vapor by the wheat population, thereby boosting productivity.

According to the performance of various traits under different sowing dates, it was found that a late sowing date would lead to an overall decrease in plant height, while spike-related traits showed a trend of increasing early but decreasing later under different environments. A previous study on the effects of late sowing dates on Xinmai23 growth period, plant height, and wheat yield showed that delayed sowing date causes gradual decreases in plant height, yield, spike numbers per hectare and the number of grains per spike while the number of grains per spike was not affected [23]. Sowing date and density show significant impacts on the plant height of Jintai 182, with PH decreasing significantly with the delay of sowing date and increasing significantly with the increase of sowing density [24]. By using five main varieties in the middle and lower reaches of the Yangtze River wheat area, the effects of different sowing date and density combinations on yield, stem morphology characteristics and lodging resistance of different varieties were compared. The results showed that with the delay of the sowing date and the increase in density, the plant height, center of gravity height and basal internode length of all varieties decreased [8]. In addition, similar phenomena have also been observed in many other different studies [25,26].

At the same time, this study also found that sowing dates changed the correlation between plant height and spike-related traits. As the sowing date was delayed, the correlation between SL and SN and the correlation between PH, ILBS and SPS changed from a nonsignificant to a significant positive correlation. Previous studies also showed that the sowing date changed the correlation of different internode lengths at the base of wheat [27]. The number of spikes and grains are key determinants of wheat yield, as they interact with grain weight to ultimately influence final yield. Modifying the sowing date can potentially alleviate the limitations imposed by the interaction between spike and grain numbers, thereby increasing wheat yield under specific conditions.

### 4.2. Stability and Comprehensive Evaluation of Wheat Plant Height Traits and Spike-Related Traits at Different Sowing Dates

Currently, there are many methods for evaluating the stability of crop target traits with no unified standards. However, the AMMI model proposed by Zhang [28] and Wu [29] is more scientific. By analyzing the genotype-environment interactions, it can intuitively demonstrate the stability of genotypes and the discrimination of environments to genotypes through biplots. This model has been widely used in soybean [30], maize [31], barley [20,21], wheat [32] and many other species.

In this study, we used the AMMI model to analyze the stability of wheat in six different combinations of sowing dates and locations. Significant differences in the stability (Di) of plant height and spike-related traits were observed among 327 varieties (lines). By sorting the value of stability parameter Di, it was found that most varieties were stable only in a single trait, among which Shannong 7859 and Zangdong 4 showed good stability in two traits. In order to reasonably judge the comprehensive stability of multiple traits of 327 varieties (lines), we conducted the comprehensive evaluation based on the entropy weight TOP-SIS model. Finally, several varieties (lines) which showed relatively strong resistance to

delayed sowing were selected. These include Xiangmai 35, Pingyang 27, Huaimai 23, and Huaimai. However, the plant height of Pingyang 27 varied from 114.67–144.75 cm in six environments despite its excellent performance. All these varieties mentioned above can be used as excellent genetic resources to improve high and stable yields.

## 5. Conclusions

Wheat is a widely cultivated cereal crop throughout the world. Sowing date has significant effects on winter wheat. This study evaluated the performance of plant height and spike-related traits under delayed sowing conditions, and significant phenotypic changes were observed. The stability of plant height and spike-related traits under different sowing dates were analyzed using the AMMI model. Based on the entropy weight TOPSIS model, several varieties, including Xiangmai 35, Pingyang 27, Huaimai 23, Huaimai 22, Emai 6, Zhenmai 12, Xiaoyan 81, Shannong 7859, Annong 1589, and Shuiyuan 86 were recommended for wheat breeders to improve stable performance under different sowing dates, which harboring good resistance to late sowing. The results of this study laid a foundation for breeding high-yield wheat varieties resistant to late sowing.

**Supplementary Materials:** The following supporting information can be downloaded at: https://www.mdpi.com/article/10.3390/agronomy13041010/s1, Table S1: Information of varieties (lines).

**Author Contributions:** S.M.O.B. performed the experiments and approved the final draft. Y.H. analyzed the data and prepared the figures and/or tables. C.L., H.X., J.Z., B.G. and F.W. performed the experiments and recorded phenotypic traits in the field. R.X. conceived and designed the experiments, authored or reviewed drafts of the paper, and approved the final draft. All authors have read and agreed to the published version of the manuscript.

**Funding:** This work was supported by the Jiangsu Province seed industry revitalization project (JBGS2021006), the Natural Science Foundation of Shanghai (22ZR1444900), and national key research and development (2017YFD0100803).

**Data Availability Statement:** Not applicable.

**Acknowledgments:** The authors would like to thank Meixue Zhou (University of Tasmania) for his help in polishing the language.

**Conflicts of Interest:** The authors declare no conflict of interest.

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
