# Peer review of "Identification of Wheat Germplasm Resistance to Late Sowing"

_agronomy, doi:10.3390/agronomy13041010_

Round 1

Reviewer 1 Report

Overview

In the study the authors aimed to analyze the effects of late sowing on wheat plant height and spike related trait in 327 wheat germplasm. I found the experiment well designed and the methods used proper for data analysis and for obtaining significant results.

The current study is on a topic of relevance and general interest to the readers of the journal. In fact, it can provide an important theoretical basis for the utilization and improvement of germplasm resources. Therefore, just some minor comments need to be addressed before final acceptance.

Minor comments

Among the 327 wheat germplasm screened, domestic and indigenous wheat varieties are included.

Landraces are known to be very valuable genetic resource for the different contemporary cereal-based farming systems. In fact, the persistence of landraces in different environments is due to their increased stability, accomplished through generations of natural and deliberate selection for valuable genes for resistance to biotic and abiotic stresses as well as for their favorable morpho-physiological and agronomic traits (Sicilia et al., 2021, Genetic and Morpho-Agronomic Characterization of Sicilian Tetraploid Wheat Germplasm).

Interestingly, when phenotypic clustering of the screened traits was performed (Section 3.4, figure 2B) many landraces clustered in the same subgroup 2. Have you noticed better performances of landraces in terms of yield traits you screened with respect to domestic varieties going through the sowing stages from the earlier to the latest? It could be interesting to show if a correlation between origin of germplasm (domestic or landrace) and yield performances exists.

·        Line 47: please substitute “which affected” with “which is affected”;

·        Line 54: please substitute “had” with “have”

·        Line 62-65: the sentence is not clear, please rephrase;

Finally, conclusions of the experiment are not described and explained.

Reviewer 2 Report

The study by Basheir et al. investigated plant height and spike-related traits in response to three sowing dates using a large set of wheat germplasms with genetic diversity at two different locations. The broad-sense heritability of those parameters was analysed. Varieties showing stable performance in those traits at different sowing dates were also identified. In general it is an interesting piece of study. English needs to be edited thoroughly.

Specific points:

For the data presented in Tables 1, 2, 3 and 4, there are too many decimals for most of the data. For example, there is no need to use decimals for plant height. Please change accordingly.

In all tables and figures, the notes need to be made in the table and figure captions to explain the full name of all parameters presented. All tables and figures  need to be able to stand alone to help reader understand all figures and tables without check all those details in the text.

No need to repeat the method used in the Results section, e.g., Lines 215-216, 228-229, 252-253, 286-287

Lines 302-305. The sentence does not make sense and is required to rephrase.

Reviewer 3 Report

It seems to us that the manuscript has no scientific value. This is a theoretical work that has no practical output, since the characteristics considered by the authors do not matter for wheat as an agricultural crop. Scientific value of the work is negligible. In fact, several morphological features are measured, and a very complex statistical analysis is performed, which does not give anything new to science. Introduction does not contain a description of the essence of such studies (using germplasm). The Name of manuscript does not reflect the essence of the work (what does resistance have to do with it, it has not been measured in any way). The methodology is not adequate (use 327 varieties to select 5 plants from them?????). The description of the climatic conditions of the area where the experiment was carried out is not given. Signs for analysis were not adequately selected (why didn’t take, for exampel, productivity, photosynthesis or photosynthetic apparatus, resistance to temperature or other stress ???). What significance do these signs (plant height, spike  length, etc.) have for science or agriculture? Neither the disign of the experiment, nor the goals, nor the results are completely clear.

Round 2

Reviewer 3 Report

We see that the authors significantly revised the manuscript and listened to the comments. The manuscript is recommended for acceptance.